# Structural Quasi-Isomerism in Au/Ag Nanoclusters

**Yifei Zhang** [1,2,†], **Kehinde Busari** [3,†] 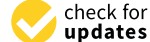, **Changhai Cao** [2,*] **and Gao Li** [3,*]

1   Institute of Catalysis for Energy and Environment, College of Chemistry and Chemical Engineering, Shenyang Normal University, Shenyang 110034, China
2   Key Laboratory of Biofuels and Biochemical Engineering, SINOPEC Dalian Research Institute of Petroleum and Petro-Chemicals, Dalian 116045, China
3   State Key Laboratory of Catalysis, Dalian Institute of Chemical Physics, Chinese Academy of Sciences, Dalian 116023, China
*   Correspondence: sdlgcch@126.com (C.C.); gaoli@dicp.ac.cn (G.L.)
†   These authors contributed equally to this work.

**Abstract:** Atomically precise metal nanoclusters are a new kind of nanomaterials that appeared in recent years; a pair of isomer nanoclusters have the same metal types, numbers of metal atoms, and surface-protected organic ligands but different metal atom arrangements. This article summarizes the structure features of isomer nanoclusters and concentrates on synthesis methods that could lead to isomer structure. The pairs of isomer inorganic nanoclusters' conversion to each other and their applications in catalyst and photoluminescence are also discussed. We found that the structure conversions are relevant to their stability. However, with the same molecule formulas, different atom arrangements significantly influence their performance in applications. Finally, the existing challenges and some personal perspectives for this novel field in the nano-science investigation are proposed. We hope this minireview can offer a reference for researchers interested in inorganic isomer nanoclusters.

**Keywords:** metal nanoclusters; structural quasi-isomerism; Au; Ag

## 1. Introduction

In chemistry, isomers mean molecules with identical formulas but various structures. The investigation of chemical compounds with isometry can be traced back to 1827 [1]. The structural differences usually cause disparate physical and chemical properties. Because of the diversity of branches of carbon chains and chirality, the structure isomerism and quasi-isomerism are universal phenomena in organic chemistry. Atomically precise metal clusters, usually smaller than 2 nm, have attracted more interest in the last decade [2–4]; by now, various kinds of metal clusters have been successfully synthesized, such as Au [5–16], Ag [17–21], Pt [21,22], Pd [23–25], Cu [26], and mixed metal clusters [27–34]. The crystal structure of metal nanoclusters can be measured using the technology of single-crystal X-ray diffraction. However, there were limited reports concerning the structure isomerism and quasi-isomerism of nanoparticles and nanoclusters due to two main reasons. Firstly, the preparation of nanoclusters at the atomic level is so complex that the prepared nanoparticles are usually polydispersed. Secondly, the different protect organic ligands could cause the difference in formulas, so even though two kinds of nanoclusters could be with the same metal atoms and numbers, they do not have an identical formula and can not be regarded as a couple of exact isomers. Research on isomers has lasted for nearly two hundred years; the reports of structural isomerism and quasi-isomerism in metal nanoclusters only appeared in the last few years [35].

Nanometal catalyst is a significant heterogeneous catalyst. In this regard, metal nanoclusters have many advantages compared to traditional metal nanoparticles because they have more activity sites (caused by the smaller size) and more selectivity (caused by the unique structure) [3]. More importantly, because of specific atomic arrangements, metal

nanoclusters could become a powerful tool for investigating catalytic reaction mechanisms. The emergence of isomers of metal nanoclusters intensifies the application for their use in the study of the catalytic reaction mechanism. Nanoclusters with different formulas usually introduce size effect and ligands effect when they are used in catalytic reactions, and isomeric structures could almost eliminate the influence of both size and ligands effects and only concentrate on structure effect in catalytic reaction progress.

Photoluminescence is another essential application for metal nanoclusters which has attracted interest in both fundamental scientific research and practical applications [36]. Because many factors could impact metal nanocluster photoluminescence, such as nanoclusters aggregation, different ligands, and Au/SR atomic ratio, to find the true influence of the metal core structure, it is crucial to keep these characteristics consistent. In the field of theoretical investigation, it is still a big challenge to reveal the influence of kernel atom structure on metal nanocluster photoluminescence.

$Au_{38}(PET)_{24}$ (PET, phenylethanethiolate) was the initially reported isomer structure in the field of clusters [8], followed by $Au_{23}(C{\equiv}CBu^t)_{15}$, $[Au_{25}(p\text{-}MBA)_{18}]^-$, et al., which have been reported in recent years [37,38]. Besides Au clusters, structural quasi-isomerism of Ag nanoclusters also has been successfully synthesized [39]. In the first section of this literature review, we will focus on the preparation of metal nanoclusters with structure isomers. The mutual transformation between organic isomer structures could happen under certain reaction conditions, and this kind of isomerization reaction could be found in the metal nanoclusters. Usually, an unstable structure could transform into a more stable one irreversibly. We will introduce it in detail in the second part of this paper.

Recently, metal nanoclusters have been widely used in various applications, such as catalysis, sensing, and photoluminescence [3]. Conventional investigation with metal nanoclusters' catalysts usually focuses on the effect of cluster size and different ligands. The appearance of metal nanoclusters with quasi-isomerism could offer an opportunity to achieve an atomic-level understanding of the relationship between metal atom arrangement and character accurately by excluding the influence of different ligands and metal atom numbers. The investigation of metal nanoclusters with isomer structures used for catalytic reactions and photoluminescence is summarized in this article.

To enrich the series of metal nanoclusters, more work must be conducted with isomer structures and extend their applications in catalytic reactions. Primarily, many common metal nanoclusters do not have corresponding isomer structures, such as $Au_{11}$, $Au_{36}$ et al. In addition, metal nanoclusters with isomer structures have been used in catalytic reactions but are so limited that more valuable reactions still need to be involved with isomer nanoclusters (biomass conversion, $CO_2RR$ et al.). Investigation of the influence of catalytic activity on isomer structures is a significant challenge. Therefore, DFT simulations will be used [1,3] to reveal the relationship between catalytic activity and the structure of metal nanoclusters. In situ characterization is a powerful tool used for revealing catalytic reaction mechanisms but was rarely reported in this reaction system. Hence, there is still much work that needs to be done with the aim of further understanding the reaction principle catalyzed by metal nanoclusters that have isomer structures. In the last part of this article, we will provide the research emphasis on structural quasi-isomerism in metal nanoclusters in the near future. Above all, this review aims to summarize the appearance and development of structural quasi-isomerism in metal clusters and their applications in catalytic reactions and photoluminescence while revealing the crucial issue in this field, giving the researchers who are interested in the relevant area fundamental material.

## 2. Synthesis of Quasi-Isomerism of Metal Nanoclusters and the Structures

Because of stable structures and smaller sizes, metal nanoclusters have been investigated in detail for many years. However, isomer structures of metal nanoclusters with single metal only appeared in the last few years. Nanocluster structural isomers are clusters with the same atom composition but different configurations. Double-metal nanoclusters could achieve isomer structure by simply changing the replacement location of the minority

metal atoms. Compared with that, the preparation of single-metal nanoclusters with isomer structures is more complicated because it requires a real difference for atomic arrangement.

Synthesis of $Au_{38}$ nanocluster was realized as early as 1993 [40]. From then on, many efforts have been carried out (such as modifying the synthesis method and introducing distinct organic ligands) to improve the yield and reveal its crystal structure. The first reported isomer nanocluster called $Au_{38T}$ was a new structure in the $Au_{38}$ family [35], and the corresponding old isomer structure was named $Au_{38Q}$. The core of $Au_{38T}$ and $Au_{38Q}$ constituted twenty-three Au atoms but differed in configurations. In $Au_{38Q}$, the core structure contains a pair of enantiomeric clusters. Each cluster has thirteen Au atoms and makes up one icosahedron. Every face of this icosahedron consists of three Au atoms, and two clusters form a rod shape by three Au atoms used in common. In $Au_{38T}$, the $Au_{23}$ core consists of one icosahedral $Au_{13}$ and one $Au_{10}$ structure unit by sharing two Au atoms; the $Au_{13}$ cluster is like a cap covered on the $Au_{10}$ cluster. Besides the core structure, the rest of the fifteen Au atoms covered the $Au_{23}$ core also in different manners; each Au atom face-capped onto the $Au_3$ faces of the $Au_{23}$ core for $Au_{38Q}$, and some faces are not capped because there were not enough Au atoms. For $Au_{38T}$, the $Au_{23}$ core was protected by six $Au_2(SR)_3$ and three Au $(SR)_2$ structure units, which can be seen in Figure 1. The $Au_{38T}$ has a more symmetrical structure than $Au_{38Q}$.

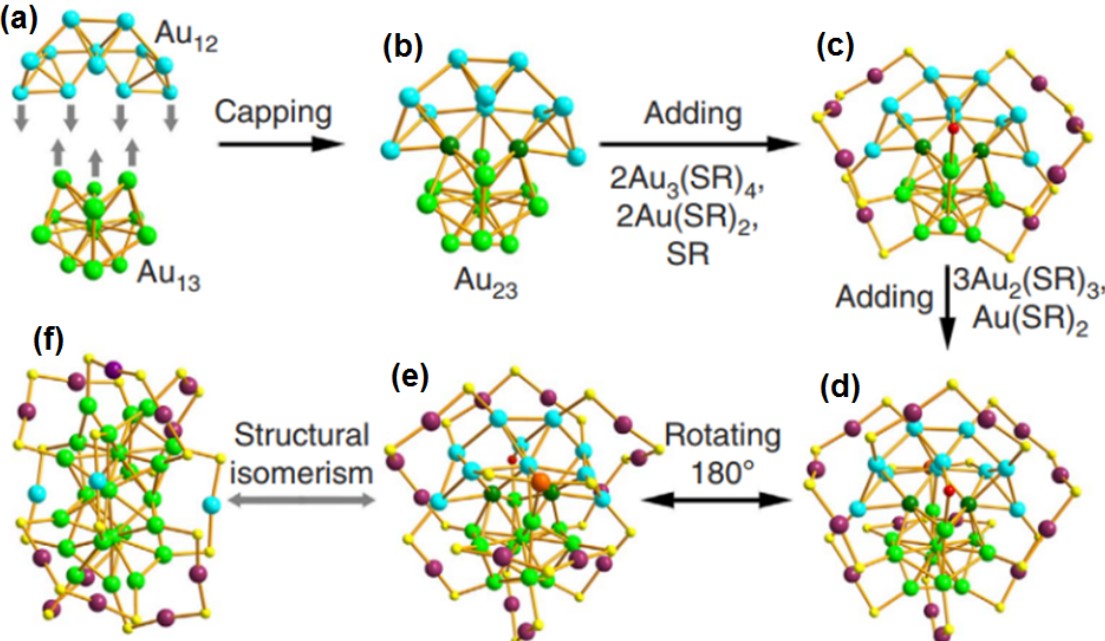

**Figure 1.** Structure details of $Au_{38T}$ and $Au_{38Q}$. (**a**) an $Au_{12}$ cap and an $Au_{13}$ icosahedral; (**b**) $Au_{23}$ core; (**c,d**) the $Au_3(SR)_4$, $Au(SR)_2$ and SR protecting the $Au_{23}$ core for $Au_{38T}$; (**e**) back view of $Au_{38T}$; (**f**) $Au_{38Q}$ structure; color code: yellow atoms, S; other colored atoms, Au. Reproduced with permission from Ref. [35], Springer 2015.

The $Au_{38T}$ was prepared by a simple one-pot method [35]. Briefly, $HAuCl_4$, TOAB, and phenylethanethiol were dissolved in $CH_2Cl_2$ and then reduced by excessive $NaBH_4$. The synthesis of $Au_{38Q}$ is more complicated. Firstly, $Au_n(SG)_m$ was prepared by the reduction method. Subsequently, $Au_{38Q}$ nanoclusters were obtained by reacting with $Au_n(SG)_m$ excess $PhC_2H_4SH$ at 80 °C. This method is commonly called etching. Considering putting various kinds of Au nanoclusters into a certain hash environment, only one type of stable structure will be left to transform into only one stable structure, and others will undergo the progress of crushing and recombination. Thus, the $Au_{38Q}$ nanoclusters were synthesized under a more rigorous situation than their isomer structure. For this reason, the structure of $Au_{38Q}$ is much more stable than $Au_{38T,}$ as further proved by the experiment.

Au$_{23}$(C≡CBu)$_{15}$ (denoted as Au$_{23}$-1 and Au$_{23}$-2) was the first reported Au nanocluster with isomer structures protected by alkynyl ligands [37]. Au$_{23}$-1 and Au$_{23}$-2 have four structure units: one Au$_{15}$ core, three V-shaped alkynyl-Au motifs, two linear "Bu-C≡C-Au-C≡C-Bu" motifs, and two bridge -C≡C-Bu ligands, and except for the Au$_{15}$ core, other units have precisely uniform structures. In different Au$_{15}$ cores, they have the same Au$_{11}$ units which were constituted by three octahedral Au$_6$ units through sharing five Au atoms. Three of the other four Au atoms have the same bonding environment in two Au$_{15}$ cores, but the fourth Au atom is not at the same position. The core of Au$_{23}$-1 has C$_2$ symmetry, which passes through the Au$_2$ atom and the center of the Au$_3$-Au$_4$ bond. The core of Au$_{23}$-2 also has C$_2$ symmetry, which passes through the Au$_5$ atom and the middle of the Au$_6$-Au$_7$ bond (Figure 2a). In Au$_{23}$-1, alkynyl bridge capping on the triangle face 1 (Au$_3$) and linear staple binding over triangle face 2. In Au$_{23}$-2, the positions of both the alkynyl bridge and linear staple changed compared with Au$_{23}$-1 because of the different structures for the Au$_{15}$ core (Figure 2b). Unlike Au$_{38T}$ and Au$_{38Q}$, which have different atom arrangement, the pair of Au$_{23}$-1 and Au$_{23}$-2 is much more similar.

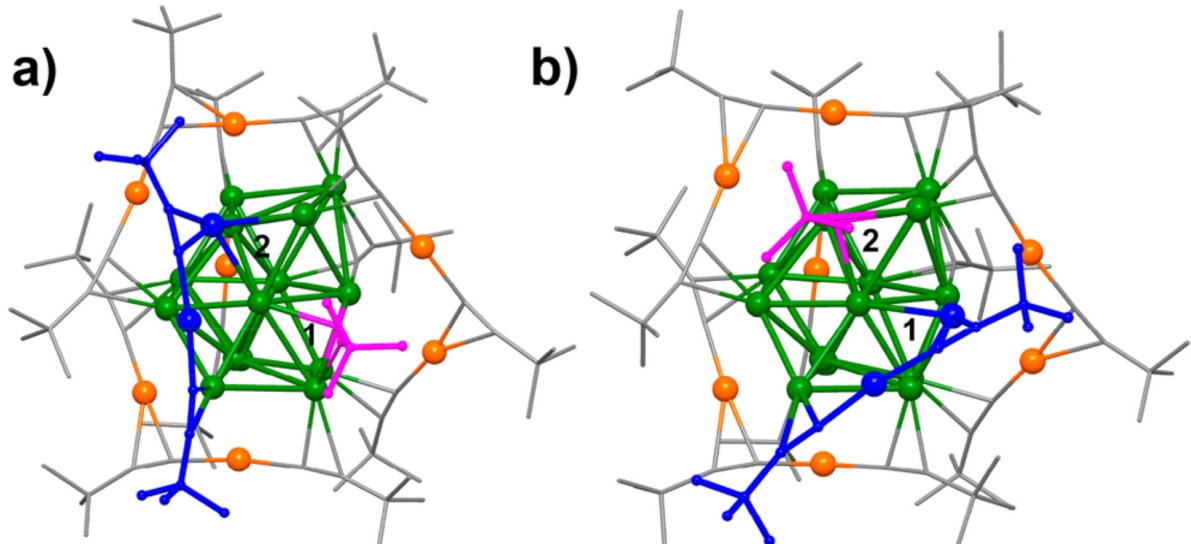

**Figure 2.** Structures of Au$_{23}$-1 (**a**) and Au$_{23}$-2 (**b**). Color code: grey atoms, C; green and orange atoms, Au; highlighted in blue, Au$_2$(C≡C-Bu)$_2$ unit; highlighted in pink, tri-bonded -C≡C-Bu. Reproduced with permission from Ref. [37]. Copyright 2020, American Chemical Society.

The synthesis of Au$_{23}$ clusters was using Me$_2$SAuCl as Au precursor mixing with HC≡CBu$^t$ and reduced by NaBH$_4$; in order to obtain Au$_{23}$-2, a certain amount of Ph$_4$P·Cl must be added after NaBH$_4$, so in this reaction system, Ph$_4$PCl was used as the inductive agent, which leads the formation of Au$_{23}$-1 to transform into Au$_{23}$-2. During the preparation process, the color change of the solution was the same for both isomer structures, similarly experiencing the transition from colorless to yellow and finally to dark brown. Because the formation of Au$_{23}$-2 needs an extra inducer, Au$_{23}$-1 is a more stable structure in the pair of isomers Au$_{23}$.

The isomer structures of face-centered-cubic (fcc) vs. non-fcc, and non-fcc-structure Au$_{42}$(TBBT)$_{26}$ nanocluster (abbreviated Au$_{42N}$) is the structural isomer of the earlier reported fcc structure Au$_{42}$(TBBT)$_{26}$ (abbreviated Au$_{42F}$) [41]. Although Au$_{42F}$ and Au$_{42N}$ have different atom arrangements, they both have regular symmetrical structures. As can be seen from Figure 3, there are thirty-four Au atoms in the core of Au$_{42F}$, and with the form of four cuboctahedras, two Au$_2$(TBBT)$_3$ dimers covered the top and bottom of the core, and four Au(TBBT)$_2$ monomers at the middle part. Au$_{42N}$ has a non-fcc core structure formed by twenty-six Au atoms. The core can be further split into three parts: the top and bottom parts have nine Au atoms, and the middle part has eight Au atoms forming

a concave quadrilateral interface. The core of $Au_{42N}$ is capped by four $Au_3(TBBT)_4$, four $Au(TBBT)_2$, and two TBBT units.

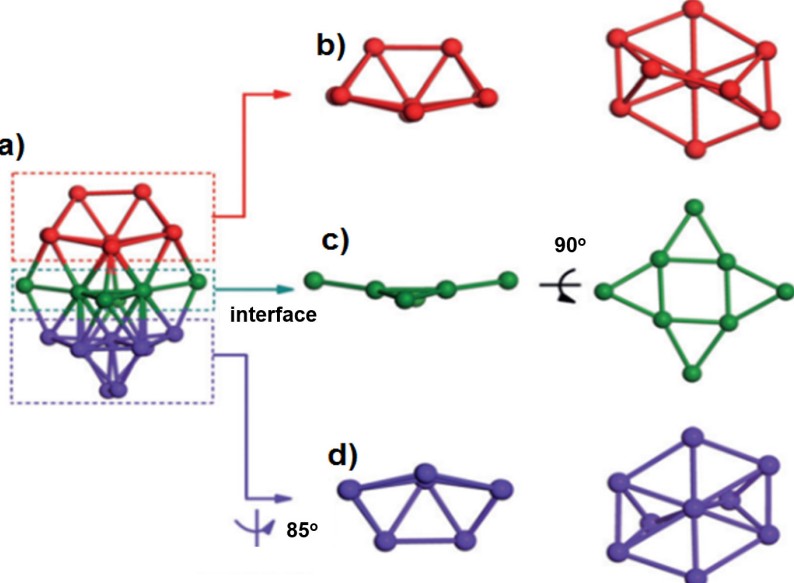

**Figure 3.** The kernel pattern of $Au_{42N}$: $Au_{26}$ kernel (**a**), view of unit 1 (**b**), interface (**c**), and unit 2 (**d**). Reproduced with permission from Ref. [36]. Copyright 2020 Wiley.

Synthesis of non-fcc-structure and fcc-structure $Au_{42}(TBBT)_{26}$ both occurred with the method of traditional $NaBH_4$ reduction, but the amount of each reagent was not the same. For example, the molar ratio of TBBT to Au was 7.71 for the synthesis of non-fcc-structure; the ratio changed to 6.1 in preparation of the fcc-structure. The most important distinction was the addition of cadmium cation in the synthesis of non-fcc-structure. Thus, without Cd, the non-fcc-structure of $Au_{42}(TBBT)_{26}$ could not be obtained. Researchers considered that Cd might influence the kinetics and thermodynamics in forming $Au_{42}$ nanoclusters, maybe creating some unstable Au/Cd intermediates or influencing the etching rate of thiol ligand, so the detailed mechanisms still need to be further investigated subsequently.

To synthesize various metal nanoclusters with isomer structures, theoretical studies have been used to predict the structures of a series of Au clusters. Xu et al. viewed $Au_4$ tetrahedron as a unit structure in building the $Au_{8n+4}(SR)_{4n+8}$ type nanoclusters, and the growth of Au kernels in thiolate-protected Au nanoclusters could be considered as the sequent addition of a basic unit. Then, the structure of $Au_{36}(DMBT)_{24}$-1D and $Au_{36}(DMBT)_{24}$-2D were predicted by a grand unified model and density functional theory calculation before being successfully synthesized [42]. Analyzed by single-crystal X-ray crystallography, both $Au_{36}(DMBT)_{24}$-1D and $Au_{36}(DMBT)_{24}$-2D can be viewed as a core with twenty Au atoms covered by eight other external staple motifs. In $Au_{36}(DMBT)_{24}$-2D, two groups of $Au_4$ units arranged in a staggered mode in the core, and the surface staple motifs can be divided into three categories: two monomeric $Au_1(SR)_2$, two trimeric $Au_3(SR)_4$, and four dimeric $Au_2(SR)_3$. In contrast to $Au_{36}(DMBT)_{24}$-2D, the helical $Au_4$ tetrahedron unit in $Au_{36}(DMBT)_{24}$-1D had the one-dimensional growth.

Different from $Au_{38}$, $Au_{23}$, and $Au_{42}$, which are pairs of isomer structures that were synthesized respectively, $Au_{36}(DMBT)_{24}$-1D and $Au_{36}(DMBT)_{24}$-2D nanoclusters were synchronously synthesized with a two-step method. Firstly, the precursor was obtained by mixing $HAuCl_4$ with organic ligands and reduced by $NaBH_4$. Subsequently, the Au precursor was further etched by DMBT for 48 h at room temperature. Then, two isomer structures were obtained by thin-layer chromatography separation and crystallized in organic solution, respectively. In this pair of isomer $Au_{36}$ nanoclusters, $Au_{36}(DMBT)_{24}$-1D is the more stable one.

$Au_{28}(CHT)_{20}$ structural isomers ($Au_{28i}$ and $Au_{28ii}$ for short) were prepared by a new quasi-antigalvanic method [43]. Similar to $Au_{36}$, the synthesis of $Au_{28}$ isomers was also conducted with a synchronous method. $Au_{23}(SC_6H_{11})_{16}$ was firstly synthesized as the precursor. Then, 1.7 equivalents of AuCHT (CHT: cyclohexanethiol) complex were mixed with an $Au_{23}$ precursor and dissolved in $CH_2Cl_2$. After stirring for 24 h at room temperature and separated by column chromatography on silica gel, $Au_{28i}$ and $Au_{28ii}$ were obtained. The structure of $Au_{28ii}$ is similar to that earlier reported: a four-tetrahedral $Au_{14}$ core is protected by four $Au_3(SR)_4$ trimers and two $Au(SR)_2$ dimers. $Au_{28i}$ has the same $Au_{14}$ core structure as $Au_{28i}$ but is covered by two $Au_3(SR)_4$ trimers and four $Au_2(SR)_3$ dimers. The structure of $Au_{28i}$ was not directly obtained from crystals of genuine $Au_{28i}$ because it cannot be formed under different conditions. After changing the protect ligand from CHT to CPT, single crystals of CPT-protected $Au_{28i}$ were obtained and analyzed by SCXD. UV-vis-NIR spectrometry confirmed that the $Au_{28i}$ structure was not varied before and after ligand exchange.

$Au_{25}(SR)_{18}$ is a common structure in the family of Au nanoclusters and has been widely used in various kinds of catalytic reactions [6,44], but the strict isomer structures with $Au_{25}(SR)_{18}$ were reported only recently [38]. As described in Figure 4, the traditional structure of $Au_{25}(SR)_{18}$ ($Au_{25R}$, Figure 4a) changed into a new isomer structure ($Au_{25G}$, Figure 4b) by collective rotation of the icosahedral $Au_{13}$ core (Figure 4b) [45]. It is worth noting that the novel structure of $Au_{25}(SR)_{18}$ was firstly predicted by molecular dynamics simulations and further confirmed by DFT. The synthesis of $Au_{25}(SR)_{18}$ could be simply described as a $NaBH_4$ reduction method, and the foundation of isomer structure was caused by the characterization of electrospray ionization ion mobility mass spectrometry. The peak observed at 74.1 ms could be ascribed to $Au_{25}(SR)_{18}$, which was predicted by the calculations earlier.

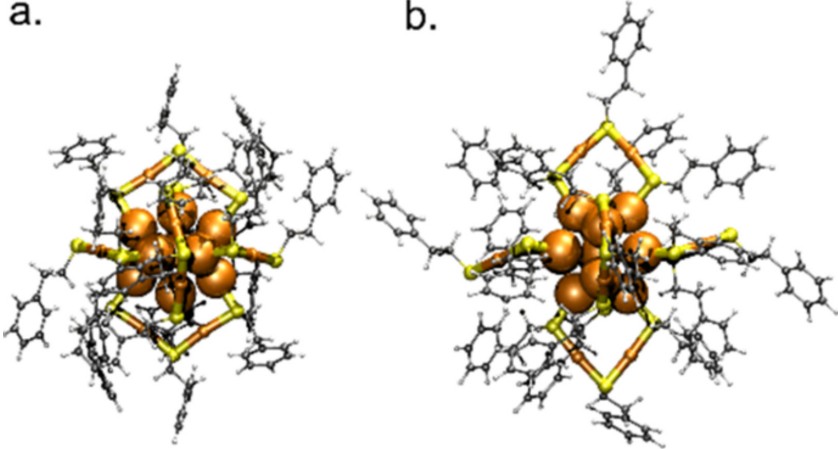

**Figure 4.** Structures of $[Au_{25}(PET)_{18}]^-$: (**a**) Experimental and (**b**) DFT. Color code: yellow atoms, S; golden atoms, Au; black atoms C; white atoms, H. Reproduced with permission from Ref. [38]. Copyright 2020, American Chemical Society.

The chiral isomer phenomenon can be widely found in the field of organic chemistry; however, inorganic compounds with chiral isomer structures were rarely reported. Mark et al. proved that introducing the chiral organic ligands in the synthesis of Ag clusters could produce the chiral effect in Ag atom arrangement [46]. The chiral ligands L/D and PL/PD lead to the formation of two pairs of enantiomeric silver clusters, $Ag_6L_6/D_6$ and $Ag_6PL_6/PD_6$ (Figure 5). Every cluster exhibits a distorted silver octahedron with six faces capped by chiral ligands that alternate in their orientation. Each Ag atom is coordinated with two sulfur atoms and one nitrogen atom from three ligands: every ligand ligates three silver atoms. In further investigation, the R/S-NYA ligands were also used in the synthesis of Ag nanoclusters with chiral isomer structures [47]. R/S-$Ag_{17}$ was prepared by a reduction reaction with triethylamine serving as a reducing agent.

Single-crystal structural analysis revealed that R/S-Ag$_{17}$ is composed of an Ag$_{17}$ core and 12 R/S-NYA ligands. The Ag$_{17}$ core can be considered as three layers with a 7-3-7 arrangement. Six edge-sharing silver triangles are located on the top and bottom. Three silver atoms form a triangle in the middle of the core structure. The method for synthesis of all Ag clusters could be simply described as dissolving silver nitrate and ligands in organic solvent and crystallization in the dark. Thus, the formation of chiral isomer Ag nanoclusters must be caused by various chiral ligands.

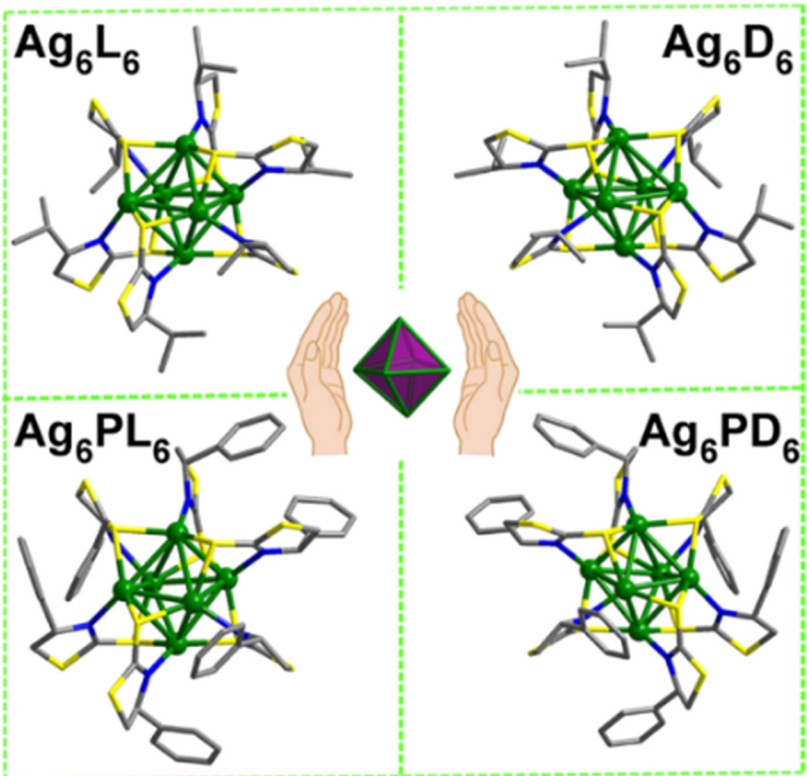

**Figure 5.** The enantiomers of Ag$_6$L$_6$/D$_6$ and Ag$_6$PL$_6$/PD$_6$. Color code: yellow atoms, S; green atoms, Ag; blue atoms, N; grey atoms, C. Reproduced with permission from Ref. [46]. Copyright 2020, Science.

In conclusion, various strict isomer structures of metal nanoclusters with single metal have been successfully synthesized by now. Among all obtained isomer clusters, the structure of Au$_{28ii}$(CHT)$_{20}$ could not be directly observed; it could only be proved indirectly, and the isomer phenomenon of Au$_{25}$(SR)$_{18}$ is detected by electrospray ionization ion mobility mass spectrometry. Every other cluster has a crystal structure. As can be seen from the results of single-crystal structural analysis, some pairs have a similar structure. Some other pairs are totally different.

Interestingly, both pairs of isomer Ag nanoclusters currently have the structure of a chiral isomer, but this kind of phenomenon could not be found in the system of isomer Au nanoclusters. The synthesis could proceed synchronously for different pairs of isomer metal nanoclusters, respectively. When the synchronous method was used, a pair of isomer structures must be separated by a chromatographic method. No matter what method was used, the obtained two isomer structures usually have different stability for Au nanoclusters. The respective synthesis method must be used for chiral isomer Ag nanoclusters because of the demand for different chiral ligands. However, distinct chirality influencing Ag nanoclusters' stability is yet to be reported.

## 3. Reversion of the Isomerism Structure

Isomerization reaction is a kind of common organic reaction that is usually induced by solid catalysts. The discovery of isomerization reaction in the field of single-metal

nanoclusters such as the isomer structures was successfully synthesized [35]. $Au_{38T}$ and $Au_{38Q}$ constituted the first reported Au nanoclusters with isomer structures. The reversion between these two structures also has been investigated in detail. At a temperature of $-10\,°C$ when dissolved in toluene, $Au_{38T}$ remained unchanged for one month. The absorption spectrum proved that, when the storage temperature was elevated to $50\,°C$, $Au_{38T}$ could transform into $Au_{38Q}$. Although many methods have been tried, the retrorse transformation could not happen, so we concluded that the structure of $Au_{38Q}$ is more stable due to synthesis with a rigorous etching method. Similar to this conversion process, the unstable $Au_{36}(DMBT)_{24}$-2D is irreversibly converted to a more stable $Au_{36}(DMBT)_{24}$-1D, which also needs a relatively high temperature of $60\,°C$. As a result, the reaction speed is much faster than $Au_{38}$ isomers (only need 2 h) [42].

Compared to $Au_{38}$ and $Au_{36}$ isomers, which need to raise the temperature for isomerization reactions, the conversion of $Au_{23}$-2 to $Au_{23}$-1 (a pair of isomer structures with a molecular formula of $Au_{23}(C{\equiv}CBu^t)_{15}$) could happen spontaneously at room temperature [37]. In the first 2 h, no change was detected by the UV-vis absorption spectrum for $Au_{23}$-2. This led to the reduction in absorption peaks of 500–600 nm to 400 nm. The absorption between 400 and 500 nm was enhanced gradually. After 7 h, the spectrum showed two distinct peaks at 412 and 598 nm, which proved the successful conversion into $Au_{23}$-1; the structure of $Au_{23}$-1 was further confirmed by MALDI-TOF-MS, and no other kinds of nanoclusters were detected. As discussed, the synthesis of $Au_{23}$-2 needs $Ph_4P{\cdot}Cl$ as the inductive agent, which finally changes $Au_{23}$-2 into $Au_{23}$-1, forming a more stable structure and proving this inductive effect to be unsustainable.

Because of the character of the oscillator, a pair of $Au_{28}(CHT)_{20}$ structural isomers ($Au_{28i}$ and $Au_{28ii}$) could repeatedly and reversibly transform into each other [43]. The conversion reaction of $Au_{28i}$ and $Au_{28ii}$ could be tuned by the solvent. Crystals of $Au_{28ii}$ dissolved in $CH_2Cl_2$ show the same absorption as $Au_{28i}$ even after several days. When a solubility-poor solvent (acetonitrile or methanol) is added, the formed crystals show the same absorption as $Au_{28ii}$, and this pathway transformation can be repeated at least 10 times. Therefore, the $Au_{28i}$ and $Au_{28ii}$ are two isomers that have been reported initially. The stability property for $Au_{28}(CHT)_{20}$ isomers is exceptional, $Au_{28i}$ is more stable in solution, but $Au_{28ii}$ prefers to exist in a crystal state.

The phenomenon of isomer structure reversibly transforming into each other also could be found in the pair of isomers $Au_{25R}$ and $Au_{25G}$ [48]. When $CTA^+$ ions were introduced to the $Au_{25R}$ solution, the interactions between $CTA^+$ ions and nanoclusters' surface changed the surface charge of nanoclusters. Zeta potential was used to monitor the state of intermolecular interactions, and the initial zeta potential was $-45$ mV, which means, under alkaline conditions, the surface of the negatively charged nanoclusters was caused by deprotonated carboxylic groups. After the addition of CTAB, zeta potential changed to $+43$ mV, indicating that the adsorbed $CTA^+$ ions are a double-layer structure. The first layer was used to balance the negative surface charge and further transferred to the positive charge by the absorption of the second layer. The color of the solution gradually changed from reddish brown to dark green by two weeks after the addition of $CTA^+$ ions, which indicated a synthesized novel species. ESI-MS revealed that these metal clusters have the same molecular formula as $Au_{25}(p\text{-MBA})_{18}$. When $Au_{25G}$ was dissolved in methanol, decoupling of $CTA^+$ ions would happen, which could be seen from the zeta potential, which changed from positive to negative. Then, the color of the solution shifted back to reddish brown after 4 h at room temperature, changing the structure back to $Au_{25R}$, which could be further proved by its absorption characteristics. Hence, the stability of $Au_{25G}$ needs persistent induction by surface-covered $CTA^+$ ions.

$Au_{25}$ isomerization reaction caused by $CTA^+$ ions will stretch the inner metallic core of Au nanoclusters and induce the isomerization process [49]. Other isomerization reactions with Au nanoclusters are all correlated with the stability of nanoclusters. Commonly, unstable structures transform into related stable structures irreversibly. Among them, $Au_{28}$ pairs are special because the isomerization reaction with $Au_{28i}$ and $Au_{28ii}$ was realized

by different stability in different phases. In all of the reported Au isomer structures, only the pair of $Au_{42}(TBBT)_{26}$ nanoclusters is not found with the isomerization reaction, which may be caused by two isomer structures having the same stability. It is worth noting that isomerization reactions are not reported with chiral isomer Ag nanoclusters. Similarly, the reason may be the same stability for chiral structures.

## 4. Catalytic Applications and Photoluminescence

Metals nanoparticles have played an essential role in industrial catalysis for a long time, to improve the catalytic activity and selectivity, revealing that the catalytic activity sites are major tasks for theoretical investigation [49]. Because traditional metal nanoparticles are usually polydisperse and have unknown surface structures, metal nanoclusters are ideal catalyst models for studying the reaction mechanism because of certain structures. By now, metal nanoclusters have been widely used in many different types of reactions such as cross-coupling, selectivity hydrogenation, et al. They could be used in correlating the catalytic properties with specific structures, and also could exhibit excellent catalytic selectivity in some cases [5]. Besides metal atoms arrangement, the kind of organic ligands covering the surface of metal clusters could also influence the catalytic activity, so different core metal structures with other ligands are not perfect catalyst models for investigation of the reaction mechanisms. In essence, the metal nanoclusters with isomer structures could solve this problem efficiently. Tian et al. used a pair of Au isomer structures, which is called $Au_{38T}$ and $Au_{38Q}$ in the reaction of 4-nitrophenol hydrogenation with $NaBH_4$ [35]. In this reaction system, 4-nitrophenol can be reduced into 4-aminophenol in 44% yield with 0.1 mol% $Au_{38T}$ catalyst for half an hour; however, no product was detected when $Au_{38Q}$ was used in the same reaction condition. The ultraviolet-visible-near-infrared spectrum revealed that the structure of $Au_{38T}$ remained unchanged even after 18 catalytic cycles. The catalyst transformed to a more stable $Au_{38Q}$ and lost all of the catalytic activity, which was observed after 21 cycles. The authors supposed that the high catalytic activity of $Au_{38T}$ may be caused by its surface that was not as protected by organic ligands as densely as $Au_{38Q}$.

Photoluminescence is an important item in the fields of both practical application and theoretical investigation. The photoluminescence of metal nanoclusters is usually influenced by many factors, and the isomers with different core metal structures will provide an excellent opportunity for investigating the kernel atom arrangement influence on the photoluminescence. Wu et al. reported that $Au_{42F}$ has stronger emission than $Au_{42N}$ (a pair of isomer structures with a molecular formula of $Au_{42}(TBBT)_{26}$), and the photoluminescence quantum yield of $Au_{42F}$ is about twice as high as $Au_{42N}$ [36]. Furthermore, density function theory calculation reveals that the charge states of total Au atoms in $Au_{42N}$ and $Au_{42F}$ are +2.678 and +2.898, respectively (electrically neutral for the whole cluster); thus, the kernel metal structures influence the charge distribution between the Au atoms and ligands. As reported earlier, the more positive charge state in the metal core prefers the charge transfer from ligands to metal part by Au-S bonds, resulting in more excellent photoluminescence performance. $Au_{42F}$ and $Au_{42N}$ have essential differences in Au atom arrangement. Ag isomer clusters only have a chiral transformation, and enantiomeric silver clusters have the same characteristics of photoluminescence [47].

By now, there is not so much literature on applications of isomer metal nanoclusters. From limited reports, we could conclude that, with the same molecular formulas, the application performance could also be influenced by core atom arrangements, except for chiral isomer, which could not impact the photoluminescence property of Ag nanoclusters. Among these reports, the discovery of specific catalytic performance with isomer Au clusters is especially important because it is the first time that directly correlates the structure and stability of metal clusters with catalyst activity. Meanwhile, it has brand new guiding significance for further investigation of catalytic reaction mechanism with nano-metal particles.

As a type of well-defined model nanocatalyst, metal nanoclusters with specific configurations have also been employed in catalytic selective oxidations [50,51]. Adding a

monodopant to a metal particle at a particular position might modify the electronic properties of these materials and alter their catalytic activity. A novel type of "pigeon–pair" cluster known as $[Au_{13}Ag_{12}(PPh_3)_{10}Cl_8]\cdot[Au_{12}Ag_{13}(PPh_3)_{10}Cl_8]^{2+}$ was synthesized by Qin et al. by doping a mono-Ag atom at the center site of $Au_{13}Ag_{12}(PPh_3)_{10}Cl_8$ nanoclusters with a rod-shape [52,53]. A variation in catalytic performance may emerge from the single-atom exchange between nanoclusters with identical structures since the electronic characteristics are disturbed [54,55]. To test the effect of a single-atom exchange on the catalytic process, $Au_{13}Ag_{12}$ and $Au_{13}Ag_{12}Au_{12}Ag_{13}$ clusters were both supported on $TiO_2$ and used in the photocatalytic conversion of ethanol. The reaction took place under UV irradiation at 30 °C. Figure 6 shows that, compared to the $Au_{13}Ag_{12}$ clusters (23%), the $Au_{13}Ag_{12}.Au_{12}Ag_{13}$ clusters had a greater conversion of ethanol, and their selectivity of ethanal (79%) was only marginally higher than that of the $Au_{13}Ag_{12}$ clusters (72%). Given how similarly distributed the products are amongst the two clusters, it is reasonable to assume that the conversion pathway and catalytic reaction mechanism should be the same for both catalysts. In essence, the identical structure of the $M_{25}$ clusters with a single atom change from Au to Ag results in a considerable variation in the catalytic activity driven by different electronic characteristics.

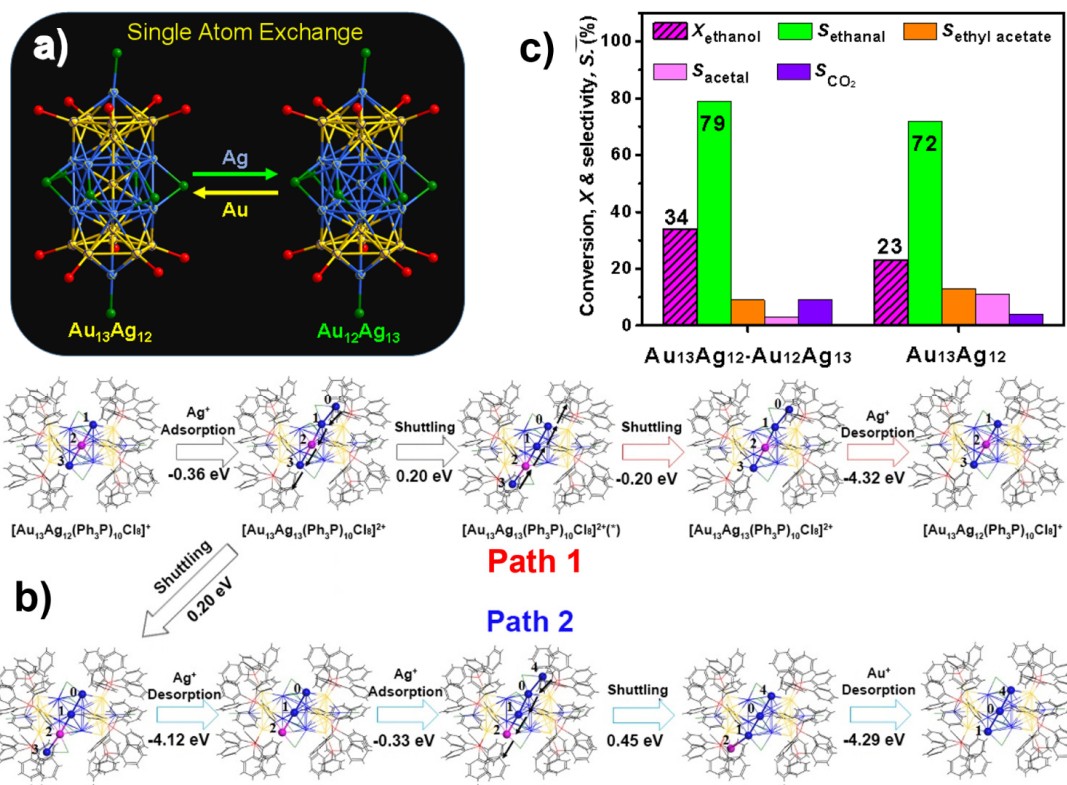

**Figure 6.** (**a**) Metal exchanging at the central site of the $M_{25}$ clusters. Color code: yellow atoms, Au; blue atoms, Ag; green atoms, Cl; red atoms, P; (**b**) pathways of the Au-Ag exchanging at the central site of $M_{25}$ clusters via DFT calculations. Color code: yellow atoms, Au; blue atoms, Ag; green atoms, Cl; grey atoms, C; white atoms, H; pink atom, exchanging atom of Au or Ag; (**c**) photocatalytic performance in the ethanol conversion. Reproduced with permission from Ref. [52]. Springer, 2022.

## 5. Conclusions and Perspectives

In this mini-review, we summarized development in recent years of the investigation of the structural quasi-isomerism with Au and Ag nanoclusters, including their metal atoms arrangements, synthesis methods, structure conversion, and applications, as listed in Table 1. The synthesis of isomer metal nanoclusters needs different craft processes, or thin-layer chromatography separation for one batch produced nanoclusters. Usually, a pair of isomer Au clusters have different stability, and the unstable one could be converted

into the more stable one irrevocably under a particular situation. Au nanoclusters with isomer structures could exhibit different characteristics in catalysis and photoluminescence, proving this novel nanomaterial to have enormous potential for both practical application and theoretical investigation. To expand species and applications for isomer clusters, much work still needs to be carried out.

**Table 1.** The synthetic process and application of these metal cluster isomers.

| Isomers | Method | Briefly Synthetic Process | Applications |
|---|---|---|---|
| $Au_{38T}$ & $Au_{38Q}$ | Respective synthesis | $Au_{38T}$ was prepared by a reduction method. $Au_{38Q}$ was obtained with an etching method | Catalytic hydrogenation of $NO_2PhOH$ |
| $Au_{28i}$ & $Au_{28ii}$ | Column chromatography | Separated by column chromatography on silica gel | - |
| $Au_{23}$-1 & $Au_{23}$-2 | $Ph_4PCl$ inductive agent | $Ph_4PCl$ as the inductive agent leads the formation of $Au_{23}$-1 to transform into $Au_{23}$-2 | - |
| $Au_{42N}$ & $Au_{42F}$ | Cd cation inductive agent | Addition of cadmium cation in the synthesis of $Au_{42N}$ | Photoluminescence |
| $Au_{36}(DMBT)_{24}$-1D & $Au_{36}(DMBT)_{24}$-2D | Column chromatography | Two isomer structures were obtained by thin-layer chromatography separation | - |
| $Ag_6L_6/D_6$ & $Ag_6PL_6/PD_6$ | Chiral organic ligands | Using the chiral ligands of L/D and PL/PD | - |
| $Au_{13}Ag_{12}$ & $Au_{12}Ag_{13}$ | Refactoring coupling | The fresh AgCl dispersed in $CH_2Cl_2$ was added in to a solution containing $Au_{13}Ag_{12}$ cluster. | Photocatalysis of ethanol |

(I) Exploitation of new structures of quasi-isomerism of metal nanoclusters.

In view of isomer structures, Au nanoclusters could be divided into three basic categories: (i) those which have been successfully synthesized and detected the single-crystal structure, (ii) one of a pair isomer structures that have not gained the specific crystal structure, and (iii) the other kinds of Au nanoclusters for which the isomer structures have yet to be discovered. By now, only a few amounts of Au nanoclusters could be classified into the first and second kinds, and in the second species, it is challenging to say isomer structures have been synthesized because the clusters do not have definite crystal structures. The reports of isomer Au clusters are limited, and the same kinds of common Au nanoclusters (such as $Au_{11}$, $Au_{13}$ et al.) have not been found with isomer structures. To enlarge the family of isomer metal clusters and further establish an isomer metal nanocluster library, it is necessary to undergo a long progress and requires a great amount of effort. As reported in works of literature, the possible methods may include thin-layer chromatography separation for known metal nanoclusters, the addition of inducing agents, and fine-tuning the synthesis formula.

(II) Exploitation of new catalytic applications of quasi-isomerism of nanoclusters.

Because of uniform particle size and specific metal atom arrangement, metal nanoclusters could exhibit excellent performance in catalytic reactions. Until now, the only report of isomer metal clusters used in the catalytic reaction is the pair of $Au_{38}$ application for 4-nitrophenol hydrogenation. As expected, the stability of nanocluster could influence their performance, and an unstable one has more activity under the same reaction system but would transform into a more stable structure during the reaction process gradually. Until recently, metal nanoparticles have been used in many kinds of valuable catalytic reactions, such as methane oxidation, biomass conversion, electrocatalytic $CO_2$ hydrogenation, and selectivity hydrogenation for unsaturated aldehyde et al. All of them require high stability of nano-catalysts, which have to be carried out under hyperthermy [56].

To use isomer metal nanoclusters in various kinds of reactions, stability is the most crucial problem that needs to be solved first. Nanoclusters supported on metallic oxide are an effective method for enhancing stability, and in some conditions, the metal oxide-supported nanoclusters catalysts could show significantly higher catalytic activity [57–60], such as the selective oxidations and hydrogenations [61,62] and photo- and electro-catalysis [63–69].

(III) DFT simulations and in situ spectroscopy on the investigation of mechanism with isomer metal clusters used in catalytic reactions.

One of the most essential objectives for the synthesis of metal nanoclusters is to achieve an atomic-level understanding of the catalytic active site. Ultrasmall nanoclusters are usually protected by organic ligands, except for particle size and metal atom arrangement. The kinds of organic ligands also play an important role in catalytic activity: metal nanoclusters with isomer structures could exclude the influence of ligands effectively, so it is a more beneficial instrument for studying the catalytic reaction mechanism. Although isomer Au nanoclusters have been used in the catalytic reactions and two different $Au_{38}$ structures exhibit distinct performance in reaction results, the specific reason is still not clear by now. DFT simulations are a powerful tool in studying the catalytic mechanism and have been wildly used in many reaction systems. Commonly, hypothetical models need to be put in DFT calculations; isomer metal nanoclusters have certain atom arrangements themselves, so they would be more suitable for taking part in DFT investigations. Compared with DFT simulations, in situ spectroscopy is a more direct method that could detect the reactants, products, and intermediates, and their combined situation with catalysts, and then reveal the detail of the reaction mechanism at the molecular level with the experimental method. In further studying the reaction mechanism catalyzed by the isomer metal clusters, DFT simulations and in situ spectroscopy play an important role in revealing the reaction progress and catalytic active site.

In short, structure quasi-isomerism with nanoclusters is a novel and worthy nanomaterial. Although the reports in this field are minimal, it is important for investigating the correlation between functions and structures. With literature published recently, we believe that more attention will be paid to the development of new isomer metal nanoclusters, which could finally become a significant part of nanometer material science.

**Author Contributions:** Investigation, Y.Z.; resources, Y.Z. and K.B.; writing—original draft preparation, Y.Z., K.B. and C.C.; writing—review and editing, K.B. and G.L.; supervision, C.C. and G.L.; project administration, C.C. and G.L. All authors have read and agreed to the published version of the manuscript.

**Funding:** This research received no external funding.

**Institutional Review Board Statement:** The study was conducted in accordance with the Declaration of Helsinki and approved by the Institutional Review Board.

**Informed Consent Statement:** Not applicable.

**Conflicts of Interest:** The authors declare no conflict of interest.

**Abbreviations**

| | |
|---|---|
| DFT | density functional theory |
| TBBT | tert-butyl benzene thiol |
| PET | phenylethanethiolate |
| p-MBA | para-mercaptobenzoic acid |
| SG | glutathione |
| R/S-NYA | *N*-((R/S)-1-(naphthalen-4-yl)ethyl)prop-2-yn-1-amine |
| DMBT | dimethyl benzene thiol |
| CHT | cyclohexanethiol |
| SCXD | single crystal X-ray diffraction |
| CTABr | cetyl trimethyl ammonium bromide |
| UV-vis-NIR | Uv-visible near infrared |
| TOAB | tetraoctyl ammonium bromide |
| ESI-MS | electrospray ionization mass spectrometry |
| MALDI-TOF-MS | matrix-assisted laser desorption/ ionization time of flight mass spectrometry |

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
