# Peer review of "Structural Quasi-Isomerism in Au/Ag Nanoclusters"

_2673-7256, doi:10.3390/photochem2040060_

Round 1

Reviewer 1 Report

The manuscript "structural isomerism in Au/Ag nanoclusters" by Zhang et al. was reviewed. The authors discussed a developing and interesting subject in nanotechnology. Structural isomerism in nanocluster, is a very resourceful chemistry in catalysis and photolumiscence applications. 

The article summarizes the structure features of isomer nanocluster and concentrate on synthetic methods that could lead to isomeric structures. From their studies, they submitted that structure conversions are linked to stability of the isomers. The mini-review is a suitable submission to the Photochem journal. The manuscript is recommended for publication. However, the following highlighted areas should be updated.

1. A table reflecting named isomers vis-a-vis applications would enhance the review 

2. The structure-property correlation and mechanism of the isomeric processes are desirable. 

3. Please check the whole length of the manuscript for grammar e.g on page 5, last line before the last paragraph; the sentence is incoherent. 

4. Please check for missing articles and punctuation in the whole lenght of the manuscript. 

Author Response

The manuscript "structural isomerism in Au/Ag nanoclusters" by Zhang et al. was reviewed. The authors discussed a developing and interesting subject in nanotechnology. Structural isomerism in nanocluster, is a very resourceful chemistry in catalysis and photolumiscence applications. 

The article summarizes the structure features of isomer nanocluster and concentrate on synthetic methods that could lead to isomeric structures. From their studies, they submitted that structure conversions are linked to stability of the isomers. The mini-review is a suitable submission to the Photochem journal. The manuscript is recommended for publication. However, the following highlighted areas should be updated.

  1. A table reflecting named isomers vis-a-vis applications would enhance the review 

A: We have changed the Table 1 to the “The synthetic process and application of these metal cluster isomers” and added the item of “Applications”

  1. The structure-property correlation and mechanism of the isomeric processes are desirable. 

A: Many thanks for Reviewer’s suggestions. We have added the structure-property correlation and mechanism of the isomeric processes.

  1. Please check the whole length of the manuscript for grammar e.g on page 5, last line before the last paragraph; the sentence is incoherent. 

A: Many thanks for Reviewer’s suggestions. We have deleted the sentence of “The two different Au36(DMBT)24 with fcc structures agree with the predicted growth method”.

  1. Please check for missing articles and punctuation in the whole lenght of the manuscript. 

A: Many thanks for Reviewer’s suggestions. We have added the missing punctuation.

Reviewer 2 Report

Isomerism of the metal clusters is one of the hot topics in the field of cluster chemistry. The authors summarized the isomerism of cluster well, however, I think if the number of metal atom had been changed, it cannot be called "isomerism". The case of Au38 is a true isomerism. And if the composition of the ligand was changed, but the number of metal atoms and ligands were not changed, it may be so-called "quasi-isomerism" (J. Am. Chem. Soc. 2016, 138, 5, 1482-1485). However, I think that if the chemical composition of metal core of the cluster was changed, we cannot call it as isomerism. I think the authors should remove the topic of Au13Ag12(PPh3)10Cl8, because the composition of metal core was changed. If it is a topic of isomerism, the pair of Au25(SR)18 and Au24Pt(SR)18 will be also isomers. It's really strange.

Author Response

Isomerism of the metal clusters is one of the hot topics in the field of cluster chemistry. The authors summarized the isomerism of cluster well, however, I think if the number of metal atom had been changed, it cannot be called "isomerism". The case of Au38 is a true isomerism. And if the composition of the ligand was changed, but the number of metal atoms and ligands were not changed, it may be so-called "quasi-isomerism" (J. Am. Chem. Soc. 2016, 138, 5, 1482-1485). However, I think that if the chemical composition of metal core of the cluster was changed, we cannot call it as isomerism. I think the authors should remove the topic of Au13Ag12(PPh3)10Cl8, because the composition of metal core was changed. If it is a topic of isomerism, the pair of Au25(SR)18 and Au24Pt(SR)18 will be also isomers. It's really strange.

A: Many thanks for Reviewer’s suggestions. We fully agree with Reviewer’s point. And we have changed the "isomerism" to "quasi-isomerism".

Round 2

Reviewer 2 Report

Thank you for considering my suggestions.

There seems to be no points to be modified.